# Autophagy Related Gene (*ATG3*) is a Key Regulator for Cell Growth, Development, and Virulence of *Fusarium oxysporum*

**DOI:** 10.3390/genes10090658

**Published:** 2019-08-28

**Authors:** A. Rehman Khalid, Xiulan Lv, Muhammad Naeem, Khalid Mehmood, Hamayun Shaheen, Pan Dong, Dan Qiu, Maozhi Ren

**Affiliations:** 1School of Life Sciences, Chongqing University, Chongqing 401331, China; 2Bioengineering College, Chongqing University, Chongqing 401331, China; 3Department of Botany, University of Azad Jammu & Kashmir, Muzaffarabad 05822, Pakistan; 4Department of Entomology, University of Poonch AJK, Rawalkot 12350, Pakistan

**Keywords:** *Fusarium oxysporum*, *ATG3* gene, autophagy, potato dry rot, disease control

## Abstract

*Fusarium oxysporum* is the most important pathogen of potatoes which causes post-harvest destructive losses and deteriorates the market value of potato tubers worldwide. Here, *F. oxysporum* was used as a host pathogen model system and it was revealed that autophagy plays a vital role as a regulator in the morphology, cellular growth, development, as well as the pathogenicity of *F. oxysporum*. Previous studies based upon identification of the gene responsible for encoding the autophagy pathway components from *F. oxysporum* have shown putative orthologs of 16 core autophagy related-*ATG* genes of yeast in the genome database which were autophagy-related and comprised of ubiquitin-like protein *atg3*. This study elucidates the molecular mechanism of the autophagy-related gene *Foatg3* in *F. oxysporum*. A deletion (∆) mutants of *F. oxysporum* (*Foatg3*∆) was generated to evaluate nuclear dynamics. As compared to wild type and *Foatg3* overexpression (OE) strains, *Foatg3∆* strains failed to show positive MDC (monodansylcadaverine) staining which revealed that *Foatg3* is compulsory for autophagy in *F. oxysporum*. A significant reduction in conidiation and hyphal growth was shown by the *Foatg3∆* strains resulting in loss of virulence on potato tubers. The hyphae of *Foatg3*∆ mutants contained two or more nuclei within one hyphal compartment while wild type hyphae were composed of uninucleate hyphal compartments. Our findings reveal that the vital significance of *Foatg3* as a key target in controlling the dry rot disease in root crops and potato tubers at the postharvest stage has immense potential of disease control and yield enhancement.

## 1. Introduction

*Fusarium oxysporum* is one of the most important fungal pathogens of tubers and root crops responsible for crop failure and yield losses across the globe [1]. The fungus attacks the crops at multiple stages initially causing the *Fusarium* wilt at planting stage followed later by the dry rot at storage stage, decreasing the crop nutrition value and causing huge market losses [2]. The yield losses were estimated between 6% to 25% annually in the field and 60% losses were recorded during storage conditions [3,4]. *F oxysporum* can be a seed-borne, soil-borne, and vascular colonization pathogen [5,6]. This fungus is ubiquitous in soil and in decaying plants material and acts as a decomposer. A study was carried out by Martius in 1842 and he found that the pathogen of potato dry rot can be a fungus, which was named a *Fusisporium*, and later on transformed to *Fusarium oxysporum* [7].

Crop tubers and roots attacked by the dry rot disease are characterized by shrinking and shriveling with lesions on the outer sides, whereas simultaneously brown to black rot damages the roots/tubers internally [8]. This pathogen penetrates into the tuber/root through wounds and then induces rot diseases, the infectious tissue becomes brown, dark red, and then form a streak, which rises up from ground level. With time course the oldest leaves become chlorotic and young leaves become flasks [1]. Thirteen species of *Fusarium* have been reported as a pathogen of dry rot [6]. From dry rotted tubers at harvest and post harvest stages, members of *Fusarium* species were reported as an important factor in causing disease [7]. During harvesting time, climatic conditions such as temperature, moisture level, as well as harvesting and handling methods, play a crucial role in the development of infection. Due to high temperature (25–30 °C) and high moisture level, tubers are most susceptible to infection as well as inefficient application of harvesting tools [9,10]. So far, for the control of *F. oxysporum*, chemical approaches have been used widely but the application of fungicides to control plant diseases causes adverse effects on the environment, soil, plants, and humans. The fumigants like methyl bromide and disinfectants like sodium hypochlorite can be harmful for young plants and cause risk to the handler and serious damage to the environment, and are not appropriate for the application [11]. Therefore, due to limitations associated with conventional, physical and chemical control methods, the resistance sources appears as a most promising technique to control phytopathogens.

Autophagy is a conserved cellular mechanism responsible for recycling and removal of cytoplasmic components including the organelles and proteins [12,13]. There are three pathways of autophagy which are existing in eukaryotes (macro, micro, and mediated autophagy), macro autophagy has been evaluated in plants, fungi, and animals [12]. Recently in yeast species *Hansenula polymorpha*, *Saccharomyces cerevisiae*, and *Pichia pastoris*, genetic analysis was performed and 42 autophagy-related (*ATG*) genes were reported. Among these 42 genes, 20 are involved in core autophagic machinery whereas the remaining 22 genes are involved in the execution of autophagy-related specific pathways triggered by a diverse array of physiological stimuli [14,15,16,17,18,19]. Recent evidence suggested that autophagy is an important factor in developmental processes of filamentous fungi such as secondary metabolism, cell differentiation, and pathogenicity [20,21]. In the current study, we have analyzed the role of *F. oxysporum Foatg3* in vegetative growth of plant pathogenic fungi, and to evaluate the importance of autophagy during different developmental stages of *F. oxysporum* by generating *Foatg3* deletion mutant and *Foatg3* overexpression strains.

## 2. Materials and Methods

### 2.1. Fungus Isloation and Culturing

*Fusarium oxysporum* strains were isolated from the infected potato tubers. These isolated strains were sensitive to the HygB at concentrations >30 mg/mL and hence were used as wild type (WT) strains in the study. The suspension of the conidia extracted from the strains was prepared by adding 50% glycerol and stored at −80 °C. We used fresh potato dextrose agar (PDA) broth medium for mycelial and conidial growth at 25 °C The *Agrobacterium tumefaciens* strain (GV3101) grown on LB media was used for the transformation of the *Fusarium oxysporum conidia* [22].

### 2.2. Fungal Transformastion

*Agrobacterium tumefaciens*-mediated transformation (ATMT) process as mentioned in the literature was performed with slight modifications. The *A. tumefaciens* GV3101 strains containing PPk2 vector were grown on PDA amended media at 28 °C The culture was mixed in a condial suspension of 107/mL at an optical density of 660 nm (OD660) reached at 0.5 with equal concentration. The fungal culture was then placed into induction medium (IM) with an optical density value of OD600 = 0.15 comprising of 200 mM acetosyringone (AS) and it was further cultured for 6 h on an orbital shaker (200 rpm) at 28 °C and 250 mM mixture was then taken out and kept on a nitrocellulose filters having a diameter of 80 mm with a 0.45 mm pore size (Whatman, Tokyo, Japan) on a co cultivation medium for a time period of 48 h. The cultured filters were further transferred to a HygB (50 mg/mL) and cefotoxime (500 mg/mL) amended selected medium to inhibit the *A. tumefaciens* growth. The randomly selected transformants were transferred to PDA media after 7 days, selected randomly after seven days, and then transferred onto HygB (50 mg/mL) enriched PDA media [23].

### 2.3. Generation of Deletion Mutants 

The fusion polymer chain reaction PCR was used to generate deletion mutant of *Foatg3* created by replacing open reading frame (ORF) of HygB resistance cassette [24]. Three primer pairs, i.e., P1/P2, P3/P4, and P5/P6 were used to amplify fragments of *Foatg3*, including 1000-bp upstream and downstream fragments, along with the 1040-bp HygB resistance cassette, from vector psilent-1. After the fusion of the *Foatg3* fragments with U-Hph-D, the digestion was performed with the restriction enzymes AsiSI and SbfI ligated with PPk2 vector (Appendix A). The PPk2-U-Hph-D recombinant plasmid of PPk2-U-Hph-D was transferred into the WT of *F. oxysporum* following the standard protocol [23]. The transformants screening was carried out by PDA media supplemented with HygB (50 µM F2du; 50 mg/mL). F-hph/R-hph PCR screening was used for the confirmation of the followed by quantitative polymerase chain reaction (qPCR) for the identification of the deletion mutants. In the experiment, F-hph/R-hph to 1021 bp was used for determining that whether the HygB is a single copy of the inserted mutant.Appendix A shows the details of the qPCR primers used.

### 2.4. Overexpression of FoATG3 Mutant Strains 

The *atg3F/R* primer was used for the amplification of the cDNA from the total RNA of *Foatg3* ORF of *F. oxysporum* which corresponded to the 7 initial ORF codons, additional cytosine, and NotI restriction site. Atg3–12 corresponded to the last 7 ORF codons with SbfI restriction site and additional cytosine for the reverse complement. Following this, the formed band was cloned to the p8GWN vector which was the plasmid template sued for the PCR amplification. The first 8 codons of AscI and SbfI restriction sites were corresponded by the *atg3*-F primer. The amplified DNA fragment was cloned into the vector F303 which resulted in the FoATG3/F303 plasmid. PCR amplification was used to obtain a Foatg3 fusion fragment (2.8 kb) controlled by the trpC promoter. Similarly, the PCR amplification resulted in the trpC terminator using the *Foatg3R* and *Foatg3F* primer pair (Appendix A), which was further utilized for the formation of the protoplast strains (Appendix A). 

### 2.5. Evaluation of Radial Growth, Conidiation, Formation, and Germination

A PDA media containing HygB (50 mg/mL) was used to analyze the radial growth and formation of the conidia. Conidia were harvested from a 10 days old culture followed by a filtration by a two layered lens paper and then re-suspension in sterile water to 1 × 10^7^ spores/mL concentration. The conidial suspensions of WT, *Foatg3∆*, and OE strains were inoculated on plates and allowed to incubate at 25 °C followed by a daily observation of the colony diameters and strain colors. Conidia (10^2^) were inoculated in PDB medium (1 mL) with continuous shaking (150 rpm) and the germination was observed at a time span of 7 h, 12 h, 21 h, 28 h, 36 h, and 48 h, respectively. The rate of conidial germination was determined by using the blood counting chamber at 12 h [24]. The data was statistically analyzed by using SPSS v.15.0 software for WindowsR (LEAD Technologies, Inc., Charlotte, NC, USA), which used student *t*-test at *p* ≤ 0.05. Fresh micro conidia (5 × 10^7^) grown on PDA medium (5 mL, 14 h, 28 °C) with 170 rpm shaking for calcoflour white (CFW) staining followed by incubation for 5 min in a dark room prior to the microscopic analysis (10 µm CFW). 

### 2.6. RNA Extraction 

The *F. oxysporum* strain was incubated in liquid PDA medium at 27 °C in the darkness for 24 h. After 24 h, the hypha was collected for RNA extraction. The RNA was extracted by using RNAiso plus (Takara, Kusatsu, Japan), as detailed in manufactured protocol in 10 mL total sample volume (5.0 µL of 2 × SYBR Premix Ex Taq, 1.0 µL of primer, 1.0 µL of cDNA, and 3 µL of distilled water deionized water). For reverse transcription reaction (cDNA) 1 µg of RNA and M-MLV reverse transcription kit (program) were used. Three biological replicates were performed in this experiment [24].

### 2.7. Quantitative RT-qPCR

Quantitative Rt-qPCR reactions were performed using a two steps methods: 95 °C for 30 s, followed by 40 cycles of 95 °C for 5 s and 60 °C for 30 s. The real time system CFX96^TM^ (Bio-Red, Hercules, CA, USA ) and Go Taq q-PCR Master Mix (Promega, Madison, WI, USA) were used for qRT-PCR analysis with specific pairs of primers as listed inAppendix A [25].

### 2.8. Analysis of Autophagy

Micro conidia (2.5 × 10^8^) inoculated on PDA medium were grown for 15 h at 28 °C to visualize the autophagy in the *F. oxysporum*. The strains were sifted to an SM medium without a N source after washing with sterile water in the presence of 4 mM PMSF (P7626, Sigma-Aldrich, St. Louis, MO, USA). The mycelia were stained in the dark for 30 min with the fluorescent dye MDC (monodansylcadaverine) with concentration of 50 mM, (Sigma, D4008) after 1 h starvation followed by washing with water and observation under epifluorescence and differential interference contrast (DIC) microscopy [26].

### 2.9. Pathogenicity Test

We used uniform sized (100–120 g) healthy tubers of potato for the inoculation essay in the current study. Tubers were initially washed to remove contamination and excessive soil followed by dipping the tubers in 0.5% sodium hypochlorite solution for 10 min and then rinsing with distilled water in 3 changes. The tubers were than sliced into small pieces and air dried. The WT, *Foatg3∆*, and *Foatg3 OE* mutant strains were incubated in the dark for 14 days on PDA medium followed by harvesting of sporangia and washing with pea broth. Inoculation of the WT, *Foatg3∆*, and *Foatg3 OE* strains was carried out on dried slices by using a 20 mL drop of sporangial suspension (1 × 10^4^ /mL) for each strain in triplicates. The inoculated tuber slices were placed in the dish on moist filter paper and then incubated for 7 days in the dark at 27 °C. The student’s *t*-test was used to measure the size of grown strains on the potato slices. Each experiment was performed in the triplicate [27].

### 2.10. Optical and Epifluorescence Microscopy

The aliquot of the fungal cells was embedded in a solution of 1% agarose blocks and was analyzed under microscope M2 Dual Cam (Carl Zeiss MicroImaging GmbH, Göttingen, Germany) equipped with the recommended filter sets for optical and epifluorescence microscopic analysis. The UV light (340–380 nm) with the filter blocks was used for staining with CFW and MDC (G 365, FT 395, and LP 420). Evolve photometric digital camera (EM512) was used along with Axiovision v.4.8 software. The imagery was further processed by using adobe Photoshop CS3 [28].

## 3. Results

### 3.1. Over Expression and Deletion of FoATG3 Mutants in F. Oxysporum

Replacement of target genes was performed in *F. oxysporum* strains to analyze the role of *Foatg3*. Polymerase chain reaction (PCR) of insertion flanking regions was used to elucidate the hygromycin resistant (HygR) transformants (Appendix A). Transformants including *Foatg3*∆#1, *Foatg3*∆#2, and *Foatg*∆#*3* successfully replacement the *Foatg3* gene and executed the expected shift in accordance to our testable hypothesis (Appendix A). qPCR was further applied for the confirmation of these putative deletion mutants (Figure 1). Co-transformation of *Foatg3* genes with different vectors successfully resulted in the over expression of *Foatg3*, which was confirmed by PCR analysis using specific primer pairs in PCR analysis (Appendix A). 

We evaluated autophagy in different strains on the nitrogen lacking medium. To achieve this target, the hyphae of these strains were stained with MDC, which is an indicator of autophagy that accumulates in vacuole in absence or presence of PMSF (phenylmethylsulfonyl fluoride), which is an inhibitor of vacuolar serine proteases [29,30]. During this study, nitrogen starved autophagy was observed, which was indicated by the presence of fluorescence dots in vacuoles and cytoplasm (Figure 2). The wild type WT and overexpressed OE *Foatg3* mutants showed positive MDC staining compared to ∆*Foatg3*. Moreover, hyphae, which were stained positively with MDC, contained degradation of nuclei while no MDC staining was observed in nitrogen starved strains of the ∆Foatg3 mutant. Hence, we concluded that Foatg3 is a basic component and necessary for starvation-induced autophagy.

### 3.2. Role of Atg3 on Conidiation and Vegetative Growth of Fusarium Oxysporum

In several fungi, autophagy-related genes effect conidial and vegetative growth [31,32]. To determine the role of atg3 we evaluated conidial formation and vegetative growth on liquid and solid media (both in rich and minimal). The radial growth of the *∆Fooatg3* mutant was slightly reduced whereas the development of aerial mycelium was significantly reduced in the high nutrition condition as compared to the wild type and *OE Foatg3* overexpressed mutant (Figure 3A,B). Meanwhile, ∆mutant showed less production of microconidia as compared to *OEFoatg3* and WT (Figure 3C). Also, in liquid culture recovered microconidia from ∆*Foatg3* mutant were less then WT and over expressed mutant (Figure 3D).

Moreover, the autophagy inducing mutants grown under severely nutrient limited conditions (SM diluted 1:1000) showed retarded growth without detectable aerial hyphae and redial growth, which was probably due to the starvation condition that encounters growth (Appendix A). Additionally, in nutrient limited conditions, the *∆Foatg3* mutant reduced conidial production in both liquid and solid medium as compared to the *OE Foatg3* mutant and wild type (Appendix A). Hence, based on these findings it is evident that autophagy is a compulsory mechanism for vegetative growth and conidial formation in *F. oxysporum*.

### 3.3. Role of Atg3 in Cellular Distribution of Nuclei in F. Oxysporum

During vegetative growth of *F. oxysporum*, nuclei divide mitotically. Thus, we hypothesized the effect of autophagy on cellular distribution of nuclei in the hyphal compartment during epical extension. The hyphal cell walls were stained with calcofluor white (CFW) chitin binding dye in order to visualize and determine the number of nuclei per compartment of hyphae. The deletion mutant (*∆Foatg3*) contained exclusively multinucleate hyphal compartments as compared to the wild type. Mostly the hyphal compartments contained two or more nuclei in the deletion mutant (Figure 4). Moreover, the mitotic pattern in the overexpressed mutant was similar to wild type. These results suggest that during vegetative growth of *F. oxysporum* autophagy contributes to the control division of nuclei in the hyphal compartment.

### 3.4. Autophagy Affects on Virulence of F. Oxysporum

To investigate the role of autophagy on the pathogenicity of *F.oxysporum*, potato tubers were inoculated with an equivalent amount of WT, *Foatg3∆*, and *Foatg3* OE spores independently. After 7 days of inoculation, aerial mycelium of different strains were observed on infected potato tubers. The colonies of the *Foatg3∆* mutant completely failed in radial growth and in formation of aerial mycelium as compared to WT and OE mutants. While wild type WT and overexpressed mutants, *OE Foatg3* increased radial growth and aerial mycelium on inoculated potato tubers (Figure 5A,B).

The disease cycle in the filamentous fungi initiates with conidia formation [33,34]. We keenly observed the infection process after inoculation in order to find out the reason for failed radial growth in the *Foatg3∆* mutant. The microscopy revealed all fungal strains to germinated conidia after 7 days; however, the *Foatg3∆* mutant was found to have fewer conidial spores as compared to both the WT and *OE Foatg3* mutant strains (Appendix A). This observed inability of the *Foatg3∆* mutant to produce the desired conidia can be seen as an important component in the mutant’s infectivity and virulence.

## 4. Discussion

Autophagy is recognized as a conserved cellular pathway evolved in eukaryotes to cope with different stresses stimuli. It is a vital protective strategy at the cellular level responsible for getting rid of unnecessary and toxic cytoplasmic materials, avoiding cellular damage due to different cellular stress, leading to changes in cellular mechanism during differentiation, and coming about as the result of turnover of organelles and proteins [35,36,37]. *F. oxysporum* is reported to exhibit an acropetal growth pattern during vegetative stages, which makes it a suitable candidate to be used as a mononucleated as well as compartmented organism in mycelial investigations [38]. Several findings indicate the presence of an effective and robust cellular mechanism to monitor the cell volume, control the compartmentalization and other stages of the cell cycle, thereby maintaining the uninucleate status of hyphal compartments during vegetative growth and fusion.

Here we adopted a reverse genetic approach to evaluate the role of autophagy-related gene *Foatg3* in this cellular maintenance process. It has been widely reported that the *F. oxysporum* genome encodes most of the known ATG proteins for nonselective macroautophagy, which is similar to other filamentous ascomycetes. In *F. oxysporum*, the gene *Foatg*8 is essential for the autophagy pathway, which participates in production of conidia, formation of appressoria and virulence of this fungus [39,40,41]. In agreement with these lines, here we found that, the *Foatg3* gene is an important component for the autophagy pathway; the autophagy-related gene *Foatg3* affects the growth and developmental process of filamentous fungus by affecting production of conidia and thus the virulence of *F. oxysporum*.

One of the key roles of autophagy is the recycling of nutrients to ensure survival of the organism in response to starvation conditions. We found nitrogen-starved autophagy in *F. oxysporum*, as reported previously in *F. graminea* [17,39]. The absence of MDC positive staining in nitrogen-starved ∆*Foatg3* mutants strongly suggests that *FoAtg3* is essential for autophagy in *F. oxysporum*. Autophagy is responsible for the internal recycling of cytosolic materials and the cellular organelles within the cells. This property facilitates the trafficking of intra cellular in the hyphal filaments subsequently enhancing the conidiophore containing aerial hyphae [18]. In agreement with this hypothesis, *∆Foatg3* mutants showed marked reduction in the growth of conidia and aerial hyphae as compared to the overexpression mutant along with the wild type strains [19,40,42]. The *Foatg3∆* mutants were also undergoing aberrant mitotic events that caused a significant increase in the fraction of hyphal compartments which contained more than one nucleus, suggesting that filamentous fungi may use nuclei as nutrient pools for supporting hyphal tip growth via autophagy. In nutrient limited conditions, the success of fungal infection relies upon the recycling and availability of the crucially important macromolecules to facilitate the hyphal growth in the host cells. It has been reported that the loss *of* the *MoATG8* gene in *M. oryzae* exponentially increases the conidial cell death by autophagy which retards appressorium arrangement ultimately leading to the loss of pathogenicity [41,43]. At par with the results in *M. oryzae*, the *CoATG8* is also attributed to be responsible for regulating the conidiation process, pattern of appressorium arrangement, as well as the pathogenicity levels in *C. orbiculare* [44,45]. The disease cycles of the filamentous fungi are initiated by the germination of spores [34]. Our findings confirm that the virulence of *∆Foatg3* mutants on potato tubers was significantly attenuated and reduced conidial production. However, reduction in conidial production can be an important factor which effects the pathogenicity of *∆Foatg3*. We found that the virulence of *Foatg3* mutants on potatoes was significantly reduced. The colonies of the *Foatg3∆* mutant completely failed in radial growth and also in formation of the arial mycelium as compared to wild type and overexpressed mutants after 7 days of inoculation. The *Foatg3∆* mutant also produced less spores as compared to wild type and overexpression. This inability of *Foatg3∆* to produce expected conidial spores is found to be the main factor responsible for its decreased infectivity. In conclusion, the present study revealed that autophagy controls formation of spores and also effects the virulence of *F. oxysporum*, which could be used for the future plant fungal disease control strategy. These results provide insights into the role of autophagy in regulation of vegetative growth and development of *F. oxysporum*. However, further studies are required to elucidate the role of *atg3* in other fungal strains and investigate the detailed functions of *Foatg3* in vegetative growth and formation of autophagosomes.

## Figures and Tables

**Figure 1 genes-10-00658-f001:**
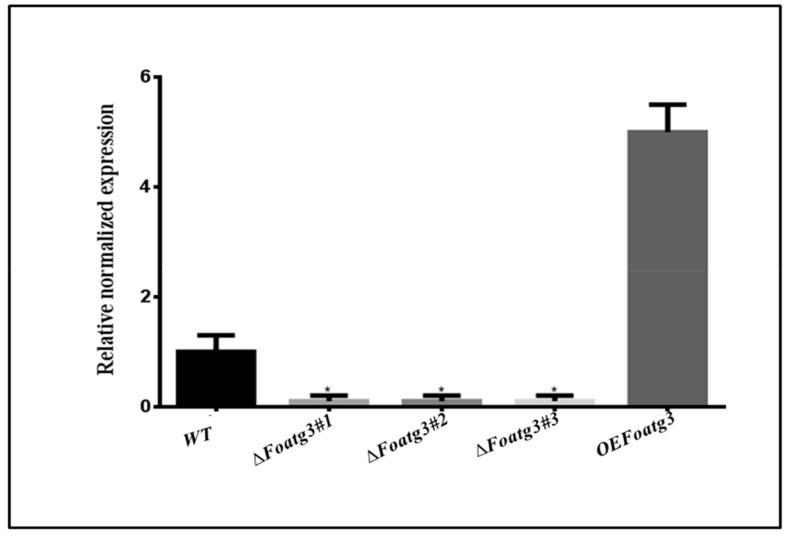
Verification of deletion and overexpression of mutants by quantitative polymerase chain reaction (qPCR). The error bars showing standard deviation. *t*-test with a * *p* value of <0.05 was applied to confirm the results statistically. WT: wild type.

**Figure 2 genes-10-00658-f002:**
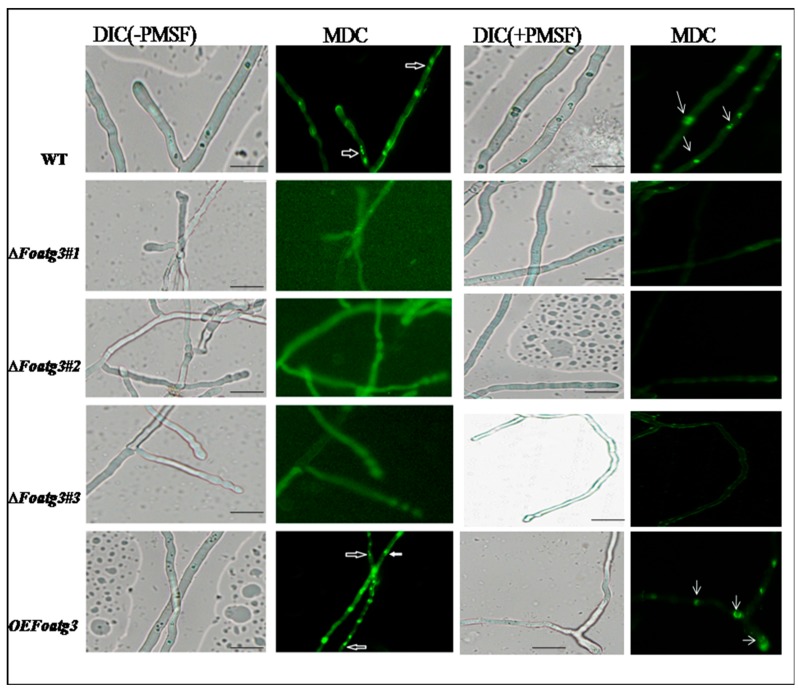
Representative phenotypes of *atg3* WT, deletion mutant, and over expression. *Foatg3* is a basic component and necessary for starvation-induced autophagy. Micro graphs showing hyphae of monodansylcadaverine (MDC) stained strains. White arrows indicating the hyphal compartment with degraded nuclei. Error Bar = 10 µm. DIC: Differential interference contrast, PMSF: phenylmethylsulfonyl fluoride.

**Figure 3 genes-10-00658-f003:**
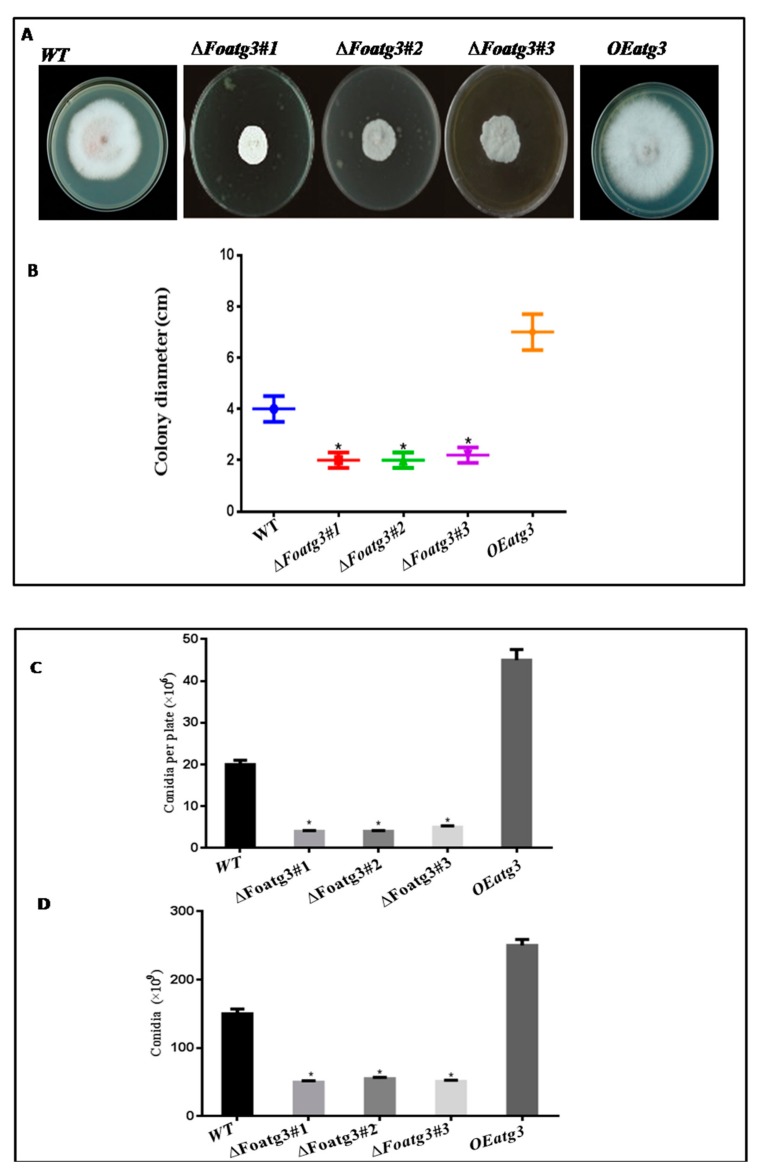
Conidia and hyphal formation were significantly reduced in the *Foatg3* mutant. (**A**) Showing fungal colony after 7 days of strain inoculation. (**B**) Graph showing comparative growth of different fungal strains. Fresh microconidia were inoculated in PDA media and then incubated at 28 °C. The diameter of the fungal colonies was measured daily for 7 days and then plotted. (**C**) Graph showing the number of microconidia grown on PDA medium at 28 °C after 7 days. (**D**) Graph showing the number of microconidia recovered after 2 days at 28 °C. The error bars on the graph indicate the standard error. *t*-test with a * *p* value <0.05 was used for statistical analysis.

**Figure 4 genes-10-00658-f004:**
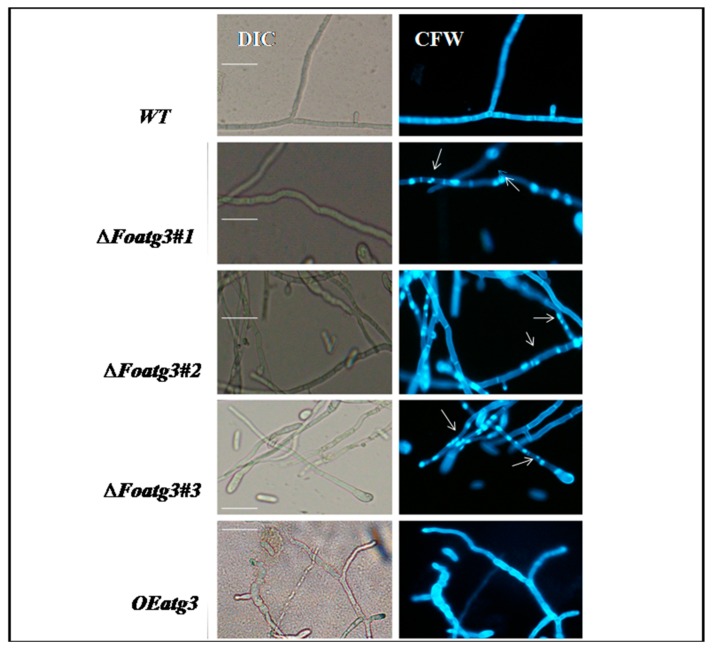
Hyphae from *Foatg3* mutants contain multinucleated compartments. Image represents hyphae of the nitrogen starvation strains which were stained with calcofluor white (CFW). Micro arrows indicating the hyphal compartment which contains more than one nucleus. Error Bar = 10 µm.

**Figure 5 genes-10-00658-f005:**
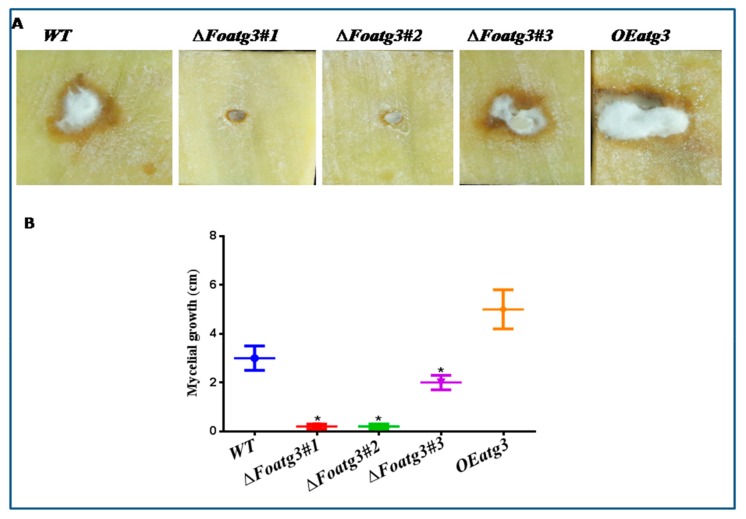
Foatg3 contributes to virulence on potato tuber slices. Mycelial growth was reduced in the *Foatg3∆* mutant. (**A**) Tubers slices were inoculated with *F. oxysporum* strains kept at room temperature and photographed after 7 days. These images represent strains growth after 7 days. (**B**) Graph represents colony formation after six days. *t*-test with a * *p* value <0.05 was performed for statistical analysis.

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
