# Peer review of "Autophagy Related Gene (ATG3) is a Key Regulator for Cell Growth, Development, and Virulence of Fusarium oxysporum"

_genes, 2019, doi:10.3390/genes10090658_

Round 1
Reviewer 1 Report
The manuscript entitled “Autophagy related gene (ATG3) is a key regulator for cell growth, development and virulence of Fusarium oxysporum” focus on the role of F. oxysporum FoATG3 at autophagy pathway especially in the development of plant pathogenic fungi, and on the evaluation of the importance of autophagy during different developmental stages of F. oxysporum by generating deletion mutant and overexpression in F. oxysporum FoATG3.
The approach used is very interesting and the methods are appropriate.
However, the manuscript revealed some points that I would like to be clarified but the authors, as follow:
- In the abstracts the authors refer to ‘The search of autophagy pathway components from F. oxysporum genome database identified 16 putative orthologs of sixteen core autophagy-related (ATG) genes of yeast, which also include the 17 ubiquitin-like protein Atg3’. This sentence seems results from this manuscript, however, I cannot find in the methods and results of the manuscript this search in the database.
- I do not understand where the genetic analysis that you describe in the sentence from L58-63 is reported.
- For expression studies it is not explained RNA extraction and cDNA synthesis. The qPCR procedure is not also explained (qPCR reagents and conditions, primer designed, reference genes selection…)
Specific comments:
L40-41: replace ‘passage of time’ by ‘time course’
L43: I think that when you write ‘a imported’ you mean ‘an important’
L44: replace ‘(…) moisture level also…’ by ‘(…) moisture level and also … ’
L46-44: rephrase sentence
L51-52: the reference [11] that support the sentence, do not refer on physical methods, but to the use of a natural disinfectant of seeds.
L74: correct the name of the transformant 3
Line 84: please explain the meaning of MDC in the first time you use the abbreviation. It is in the abstract but it has also to be explained in the body text. Also, in L85 the meaning of PMSF. Please check all abbreviations.
Line 253-254: please clarify from table S1 which primers were used.
L258: I think that you mean ‘TAIL-PCR’ protocol instead of ‘AIL-PCR’ protocol. Also ‘was followed’ instead of ‘will be followed’.
L259-260: Where are indicated RB and LB primers? Which ones from publication [45]? Also I do not understand what you took from reference [46].
Line 286: I do not identify the refereed primers in table S1. Please check if the names of the primers across through the manuscript correspond to the ones identified in Table S1.
Replace qRT-PCR by qPCR.
Please use only Fusarium oxysporum only the first time and then F. oxysporum always in italics.
In the legends of the figures please include what all abbreviation means.
Fig. S2: Please indicate the size of the amplicons in the gel. Also, you refer to A and B in the legend but you have three images.
Please check with detail the materials and methods section.
Although English is not my mother-tongue, I made some English corrections, but I think that the manuscript should undergo extensive English editing.
Author Response
Response to Reviewer #1:
We would like to thank the reviewer for careful and thorough reading of this manuscript and for
the thoughtful comments and constructive suggestions, which help to improve the quality of this
manuscript. Our response follows (the reviewer’s comments are in italics).
General Comments
The manuscript entitled “Autophagy related gene (ATG3) is a key regulator for cell growth, development and virulence of Fusarium oxysporum” focus on the role of F. oxysporum FoATG3 at autophagy pathway especially in the development of plant pathogenic fungi, and on the evaluation of the importance of autophagy during different developmental stages of F. oxysporum by generating deletion mutant and overexpression in F. oxysporum Foatg3.
The approach used is very interesting and the methods are appropriate.
However, the manuscript revealed some points that I would like to be clarified but the authors, as follow:
In the abstracts the authors refer to ‘The search of autophagy pathway components from F. oxysporum genome database identified 16 putative orthologs of sixteen core autophagy-related (ATG) genes of yeast, which also include the 17 ubiquitin-like protein Atg3’. This sentence seems results from this manuscript, however, I cannot find in the methods and results of the manuscript this search in the database.
We appreciate the positive feedback from the reviewer.
1) With regards to question about abstract as we noted in our response to Reviewer #1. According to kind suggestions of reviewer we have reviewed carefully the entire manuscript and have removed redundancies, as shown in the revised manuscript.
Reply
Corrections has been made according to kind suggestion
2) I do not understand where the genetic analysis that you describe in the sentence from L58-63 is reported.
Reply
Correction has been made according to kind suggestions
3) For expression studies it is not explained RNA extraction and cDNA synthesis. The qPCR procedure is not also explained (qPCR reagents and conditions, primer designed, reference genes selection…)
Reply
Correction has been made according to kind suggestions.
Minor Comments:
1) L40-41: replace ‘passage of time’ by ‘time course’
Reply:
As suggested by reviewer correction has been made.
2) L43: I think that when you write ‘a imported’ you mean ‘an important’
Reply:
Corrections has been made as suggested.
3)L44: replace ‘(…) moisture level also…’ by ‘(…) moisture level and also … ’
Reply
Corrections has been made as suggested.
4) L46-44: rephrase sentence
Reply
Correction has been made as suggested
5) L51-52: the reference [11] that support the sentence, do not refer on physical methods, but to the use of a natural disinfectant of seeds.
Reply
Correction has been made as suggested.
6) L74: correct the name of the transformant 3
Reply
Correction has been made
7) Line 84: please explain the meaning of MDC in the first time you use the abbreviation. It is in the abstract but it has also to be explained in the body text. Also, in L85 the meaning of PMSF. Please check all abbreviations.
Reply
Correction has been made according to kind suggestions.
8) Line 253-254: please clarify from table S1 which primers were used.
Reply
Correction has been made as suggested.
9) L258: I think that you mean ‘TAIL-PCR’ protocol instead of ‘AIL-PCR’ protocol. Also ‘was followed’ instead of ‘will be followed’.
Reply
Correction has been made
10) L259-260: Where are indicated RB and LB primers? Which ones from publication [45]? Also I do not understand what you took from reference [46].
Reply
We mentioned this refer because we followed procedure of this paper. However text has been revised carefully as suggested.
11) Line 286: I do not identify the refereed primers in table S1. Please check if the names of the primers across through the manuscript correspond to the ones identified in Table S1.
Reply
Changes has been made according to kind suggestions.
12) Replace qRT-PCR by qPCR.
Reply
Please use only Fusarium oxysporum only the first time and then F. oxysporum always in italics.
13) In the legends of the figures please include what all abbreviation means.
Reply
Corrections has been made as suggested
14) Fig. S2: Please indicate the size of the amplicons in the gel. Also, you refer to A and B in the legend but you have three images.
Reply
Corrections has been made
15) Please check with detail the materials and methods section.
Although English is not my mother-tongue, I made some English corrections, but I think that the manuscript should undergo extensive English editing.
Reply
Whole manuscript has been revised with extreme care and text has been revised carefully according to kind suggestions.

Reviewer 2 Report
Author demonstrated autophagy pathway components from F. oxysporum genome database identified putative orthologs of sixteen core autophagy-related (ATG) genes of yeast, which also include the ubiquitin-like protein Atg3. The deletion (∆) mutants of F. oxysporum (Foatg3∆) were generated to evaluate nuclear dynamics. As compared to wild type and Foatg3 OE (over expression) strains, Foatg3∆ strains did not show positive MDC (monodansylcadaverine) staining which revealed that Foatg3 is compulsory for autophagy in F. oxysporum. The Foatg3∆ strains showed reduction in hyphal growth and conidiation and lost virulence on potato tubers. The hyphae of Foatg3∆ mutants contained two or more nuclei within one hyphal compartment while wild type hyphae were composed of uninucleate hyphal compartments. FoATG3 may be a key target for the control of dry rot disease in tuber and root crops during postharvest stage. Although the overall interest and visibility of this work, some aspects should still be considered to improve the quality and objectiveness of this work.
The same author published paper entitled “Role of Autophagy-Related Gene atg22 in Developmental Process and Virulence of Fusarium oxysporum” in Genes, 2019. Some novelty needed from previous publication of same authors.
· Abstract is not clear. Abstract needs clear background, objectives, methods, results and conclusion. But the present form of abstract is not clear.
· Background of the study should be made to very clear.
· Introduction” is inappropriate and some statements are given without reference citation so rewrite it only within 1 and half page as: First of all present the background studies about the topic in a manner that set a foundation to understand the research problem with proper reference citations.
In “Materials and Methods” majority portion is given without any reference citation.
Rewrite the “Results” with proper subheadings and avoid the extra general and unnecessary information in it.
Discussion part is very poor presentation.
Conclusion” of the study is lengthy. Summarize it in 5-6 lines and provide only the core outcomes of study in it.
· Overall, this manuscript written is very poor.
English of the MS needs to be critically improved.
Minor comments
· Line 21: F. oxysporum to change italic. Line 35: Fusisporium. In many places microorganism name is not italic. Foatg3 gene also should be italic.
· Author needs to change.
· In many places in text F.oxysporum to change F. oxysporum.
Author Response
Response to Reviewer #2:
We would like to thank the reviewer for careful and thorough reading of this manuscript and for
the thoughtful comments and constructive suggestions, which help to improve the quality of this
manuscript. Our response follows (the reviewer’s comments are in italics).
General comments:
Author demonstrated autophagy pathway components from F. oxysporum genome database identified putative orthologs of sixteen core autophagy-related (ATG) genes of yeast, which also include the ubiquitin-like protein Atg3. The deletion (∆) mutants of F. oxysporum (Foatg3∆) were generated to evaluate nuclear dynamics. As compared to wild type and Foatg3 OE (over expression) strains, Foatg3∆ strains did not show positive MDC (monodansylcadaverine) staining which revealed that Foatg3 is compulsory for autophagy in F. oxysporum. The Foatg3∆ strains showed reduction in hyphal growth and conidiation and lost virulence on potato tubers. The hyphae of Foatg3∆ mutants contained two or more nuclei within one hyphal compartment while wild type hyphae were composed of uninucleate hyphal compartments. FoATG3 may be a key target for the control of dry rot disease in tuber and root crops during postharvest stage. Although the overall interest and visibility of this work, some aspects should still be considered to improve the quality and objectiveness of this work.
1) The same author published paper entitled “Role of Autophagy-Related Gene atg22 in Developmental Process and Virulence of Fusarium oxysporum” in Genes, 2019. Some novelty needed from previous publication of same authors.
Reply
Thank you very much for carefully revision of manuscript, with regards to your question, in previous studies we emphasized on role of atg22 in autophagosome formation while in present study we discussed other parameters like role of autophagy in nuclear dynamics and vegetative growth.
2) Abstract is not clear. Abstract needs clear background, objectives, methods, results and conclusion. But the present form of abstract is not clear.
Reply
Changes has been made as suggested
3) Background of the study should be made to very clear.
Reply
Corrections has been made as suggested.
4) Introduction” is inappropriate and some statements are given without reference citation so rewrite it only within 1 and half page as: First of all present the background studies about the topic in a manner that set a foundation to understand the research problem with proper reference citations.
In “Materials and Methods” majority portion is given without any reference citation.
Rewrite the “Results” with proper subheadings and avoid the extra general and unnecessary information in it.
Discussion part is very poor presentation.
Conclusion” of the study is lengthy. Summarize it in 5-6 lines and provide only the core outcomes of study in it.
Overall, this manuscript written is very poor.
English of the MS needs to be critically improved.
Reply
We appreciate the positive feedback from the reviewer. With regards to review writing and punctuation of manuscript, as we noted in our response to Reviewer. According to kind suggestions of reviewer we have reviewed carefully the Introduction, Material & Method, Results and discussion and have removed redundancies, as shown in the revised manuscript.
Minor comments
· Line 21: F. oxysporum to change italic. Line 35: Fusisporium. In many places microorganism name is not italic. Foatg3 gene also should be italic.
· Author needs to change.
· In many places in text F.oxysporum to change F. oxysporum.
Reply
Corrections has been made according to kind suggestions.

Round 2
Reviewer 1 Report
The authors reply to almost all the raised point. Please still consider:
Regarding my previous point
3) For expression studies it is not explained RNA extraction and cDNA synthesis. The qPCR procedure is not also explained (qPCR reagents and conditions, primer designed, reference genes selection…)
I cannot see where the authors include this information or a reference from where this information was taken.
Minor points:
Line 44: replace ‘imported’ by ‘important’
Line 46-48 in green : Why is it in green? Is to be removed? I think that it can be removed.
Fig. S2: Please indicate the size of the amplicons in left side of the ladder.
Author Response
Response to Reviewer #1:
We would like to thank the reviewer for careful and thorough reading of this manuscript and for
the thoughtful comments and constructive suggestions, which help to improve the quality of this
manuscript. Our response follows (the reviewer’s comments are in italics).
Reviewer # 1
3) For expression studies it is not explained RNA extraction and cDNA synthesis. The qPCR procedure is not also explained (qPCR reagents and conditions, primer designed, reference genes selection…)
Reply
Changes has been made according to kind suggestions
I cannot see where the authors include this information or a reference from where this information was taken.
Reply
Changes has been as suggested
Minor points:
Line 44: replace ‘imported’ by ‘important’
Line 46-48 in green : Why is it in green? Is to be removed? I think that it can be removed.
Fig. S2: Please indicate the size of the amplicons in left side of the ladder.
Reply
All minor changes has been made according to kind suggestions

Reviewer 2 Report
Requested corrections were carried out by authors. Statistics was not clear in figures graph.
Author Response
Response to Reviewer #2:
We would like to thank the reviewer for careful and thorough reading of this manuscript and for
the thoughtful comments and constructive suggestions, which help to improve the quality of this
manuscript. Our response follows (the reviewer’s comments are in italics).
Reviewer # 2
Requested corrections were carried out by authors. Statistics was not clear in figures graph
Reply
The data was statistically analyzed by using student T-test at P ≤ 0.05. It is mentioned in methodology and once again checked graphs and figs according to kind suggestions and changes has been made.
